# Research on Power Service Route Planning Scheme Based on SDN Architecture and Reinforcement Learning Algorithm

Xinquan Lv [1], Yongjing Wei [1], Kai Ma [1], Xiaolong Liu [2], Chao Sun [1], Youxiang Zhu [1] and Piming Ma [2,*]

1 Information & Telecommunications Company, State Grid Shandong Electric Power Company, Jinan 250013, China; lv521380@163.com (X.L.); 13791138981@163.com (Y.W.); 15695295716@163.com (K.M.); sunchaosdu@163.com (C.S.); zhuxyz@163.com (Y.Z.)
2 Shandong Provincial Key Laboratory of Wireless Communication Technologies, The School of Information Science and Engineering, Shandong University, Qingdao 266237, China; 202112687@mail.sdu.edu.cn
* Correspondence: mapiming@sdu.edu.cn

**Abstract:** The power communication network carries various power services to ensure the safe operation of the power network, among which, the relay protection service is the most important service. Reasonable planning of the service route can improve the effectiveness and reliability of data transmission in the power communication network, thereby ensuring the reliable operation of the power grid. This paper constructs a route planning architecture for the power communication network based on a software-defined network. On this basis, parameters such as the power service and network-carrying service status are defined. With the goal of minimizing network risk variance and considering link bandwidth utilization and overload constraints of relay protection services, a service route allocation problem has been raised. To solve this problem, a power service route planning scheme based on a reinforcement learning algorithm is proposed. This algorithm uses the state–action–reward–state–action (SARSA) algorithm to complete service route planning. The simulation results show that using the route planning scheme proposed in this paper can avoid the overload of relay protection services, reduce network risk variance, and effectively balance network risk.

**Keywords:** power communication network; overload; dual route; software-defined network; reinforcement learning

## 1. Introduction

With the development of communication technology and computer technology, traditional power grids are evolving toward smart grids with higher reliability, sustainability, and flexibility [1–3]. The power communication network is an important component of the smart grid, and the backbone network consists of exchange nodes and fiber optic links, carrying out the services of power grid production and operation. The transmission and exchange of service data are closely related to the safe operation of the power grid [4,5]. In the pursuit of high-quality power grid development, the variety and number of services continue to grow, and it is necessary to ensure the reliability and effectiveness of power communication network service transmission [6]. Among them, an important method for ensuring the safe and reliable operation of the power communication network is to allocate routes reasonably for power services [7].

At present, in terms of network technology, software-defined networking (SDN) decouples the data plane from the control plane, simplifies network management, and is currently a popular network technology [8]. In addition, reinforcement learning (RL) and deep learning (DL) are developing rapidly and have shown strong capabilities in various fields. DL combines low-level features through multi-layer network structures and nonlinear transformations to form abstract and easily distinguishable high-level representations,

in order to discover distributed feature representations of data. DL focuses on the perceptions and expressions of things, and generally requires a large amount of training data. RL maximizes the cumulative reward value that the agent receives from the environment to learn the optimal strategy for achieving goals, with a focus on learning problem-solving strategies [9]. RL is more suitable for route planning problems.

For research on route planning schemes for power services, the main focus is currently on reducing network risks, and there is a lack of consideration for relay protection service overload (RPSO). A relay protection service is the most important service carried out by the power communication network and the most important factor for the safe and stable operation of the power grid. To ensure the safe transmission of the relay protection service, route planning is constrained by overload; that is, the number of relay protection services carried by an optical cable link in the network must not exceed the overload threshold. At the same time, relay protection services require dual route planning. When the working route fails, it quickly switches to the protection route to ensure the transmission of service data. Therefore, in response to the overload constraint and dual route planning of relay protection services, this paper proposes a route planning scheme based on the state–action–reward–state–action (SARSA) algorithm. The simulation experiment verified the effectiveness of the route planning scheme proposed in this paper. The main contributions and novelty of this paper can be summarized as follows:

1.  We establish a power communication network service route planning architecture based on SDN and define power service and network state parameters.
2.  We aim to reduce the variance of network risk and propose a route planning scheme based on the SARSA algorithm for service routes while satisfying the conditions of link bandwidth and RPSO.
3.  Assuming that power services arrive in chronological order, we provide a route planning process for multiple services.

## 2. Related Work

A large number of research studies have been published on the route planning of power services. Reference [10] calculates the importance of links based on the availability of links in the power communication network and the service routes carried, and then analyzes the reliability of each service route. References [11,12] divide the power communication network into the physical link layer, network topology layer, and service layer. Based on the three-layer topology structure and its relationships, the reliability of switching equipment and fiber optic cable links is analyzed, and a risk assessment model for the service route is established. On this basis, many scholars have established different service route planning models that consider the risk or risk balance of the power communication network, but their solution methods are different.

One type is to use algorithms from graph theory, including the Dijkstra algorithm and k shortest path (KSP) algorithm. In reference [13], the authors consider the risk and load of the link, and use the weighted average of the two as the link weight, ignoring the weight of the exchange node. The Dijkstra algorithm is used to solve the problem. Similarly, in reference [14], the authors take the risk of fiber optic links and switching nodes as their weights and use the Dijkstra algorithm to directly obtain the route with the lowest risk value as the service route. Reference [15] uses the KSP algorithm to search for multiple reachable routes for services and selects the route with the lowest network risk balance as the service route. Reference [16] considers the situation of route recovery after communication network failures. For services that cannot be transmitted due to communication network failures, the KSP algorithm obtains k candidate paths based on the link bandwidth and transmission delay, and selects the path that meets bandwidth requirements as the recovery route. Reference [17] considers the risks of nodes and links, calculates the shortest path for each service based on the intermediary centrality theorem, and modifies the service route multiple times to improve the risk balance of the network.

Another type is to use heuristic algorithms such as the genetic algorithm. In reference [18], the author weighs the risk balance sum, load pressure, and average service delay as the objective function for route planning, and uses an improved genetic algorithm to solve it. Reference [19] considers the multi-objective optimization problem of establishing risk balance and service delay, both of which are solved using the NSGA-II algorithm. Reference [20] constructs a link importance evaluation algorithm for both working and backup routes based on the SDN architecture, and optimizes network risk using a genetic algorithm for service route planning. Reference [21] constructs a communication vulnerability index based on service distribution and network attacks to describe the risk of service transmission, and optimizes the route through an improved fast genetic algorithm.

The last type is to use the RL algorithm, such as the Q-learning algorithm. Reference [22] proposes a route optimization algorithm based on reinforcement learning for low latency services in power communication networks, with the sum of data flow delays as the reward value. Reference [23] introduces the edge risk value weight and node risk value weight to improve the existing edge risk value and node risk value indicators, and proposes a Q-learning algorithm-based service route planning algorithm for the power communication SDH optical transmission network. Reference [24] defines the ratio of the joint importance of a service to its path reliability as the joint importance reliability value, and proposes a Q-learning-based route planning algorithm with the standard deviation of this value as the objective optimization function. Reference [25] proposes a route optimization algorithm based on reinforcement learning and multi-constraint fusion for the power communication OTN transmission network services. In the simulation, this paper compares the performance of the Q-learning algorithm and the SARSA algorithm. The largest difference between the SARSA algorithm and the Q-learning algorithm is that the Q-learning algorithm uses the maximum value of the value function for iteration, while the SARSA algorithm uses the actual Q-value for iteration. Compared to the Q-learning algorithm, the SARSA algorithm tends to converge more easily. Therefore, we adopt the SARSA algorithm to solve the route planning problem.

## 3. System Model

The power communication network route planning architecture based on SDN is shown in Figure 1, which includes the data plane, control plane, and calculation plane. The data plane mainly refers to the power communication network. In the power communication network, service data are transmitted from the source node to the destination node through route. The control plane mainly includes the SDN controller. The SDN controller controls the reception of service information, such as the service source node, service destination node, etc., and obtains the current network status, including the node-carrying service situation and link-carrying service situation. Furthermore, the SDN controller takes the service information and network status as inputs and calls the route algorithm of the calculation plane to plan the route for the current service. After obtaining the service route, the SDN controller distributes the service route to the power communication network and updates the network status to complete the service route planning. Below is a detailed introduction to the status of the power communication network, power service, and network-carrying service status.

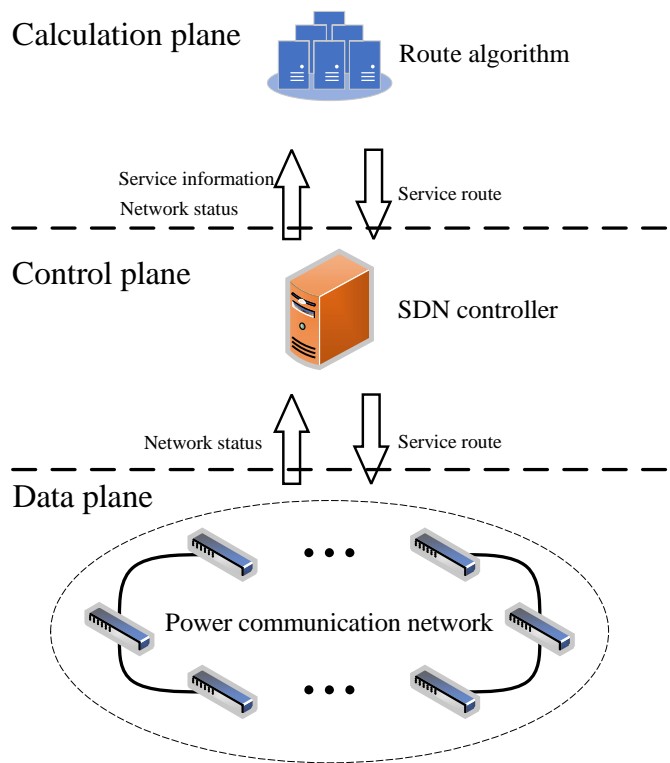

**Figure 1.** Architectural diagram of power communication network route planning based on SDN.

### 3.1. Power Communication Network

The power communication network consists of $N$ switching nodes and $M$ fiber optic links, which can be represented as $G = (V, E)$. Among them, $V = \{v_i | i \in \Lambda\}$, $\Lambda = \{1, 2, ..., N\}$ represents the set of switching nodes, and $E$ represents the set of optical cable links. The connectivity of switching nodes can be represented by the adjacency matrix $A$. If there is an optical cable link between node $v_i$ and node $v_h$, then the element $A(i, j) = 1$; otherwise, $A(i, j) = 0$. Therefore, the adjacency matrix is a symmetric matrix, and $A(i, j) = 1$ and $A(j, i) = 1$ represent the same optical cable link. We define the fiber optic cable link as $e_{ij}$ to avoid duplication and meet $i < j$. Therefore, the fiber optic cable link set $E = \{e_{ij} | A(i, j) = 1, i < j, i \in \Lambda, j \in \Lambda\}$ has $M$ elements.

In the power communication network, power service data are generally sent out from the source node, pass through multiple switching nodes and fiber optic links, and finally reach the destination node. Considering the issues of fiber optic cable bandwidth and network risk, we further define the relevant parameters of switching nodes and fiber optic cable links.

In the power communication network, the availability of switching nodes will be reduced due to factors such as their service life and operating environment. Assuming the availability of exchange node $v_i \in V$ is $u_v(v_i)$, its failure efficiency is

$$r_v(v_i) = 1 - u_v(v_i). \tag{1}$$

For the fiber optic cable link $e_{ij} \in E$, its length and bandwidth capacity are expressed as $l(e_{ij})$ and $f(e_{ij})$, respectively. Given the single-core bandwidth capacity of a fiber optic cable link, the bandwidth capacity of the link is directly proportional to the number of fiber optic cable cores. At the same time, the fiber optic cable link will set a bandwidth utilization threshold to reserve a portion of the bandwidth for emergency situations, assuming the bandwidth availability of link $e_{ij}$ is $\eta_m(e_{ij})$. In addition, during the use of optical cables, the availability of cable links may be reduced due to natural or human factors. Natural factors include a long service life, environmental erosion, and geological disaster damage. Human factors include malicious human destruction and engineering development damage. As-

suming that the unit length availability of the fiber optic cable link $e_{ij}$ is $u_{\mathrm{e}}(e_{ij})$, its failure efficiency is expressed as

$$r_{\mathrm{e}}(e_{ij}) = 1 - u_{\mathrm{e}}{}^{l(e_{ij})}(e_{ij}). \tag{2}$$

### 3.2. Power Service

There are various types of power services, including relay protection services and stable control system services. The bandwidth requirements for different services vary, and their importance for the normal operation of the power grid also differs. Assuming the number of types of electricity services is denoted as $K$, the importance of each type of service can be calculated using the analytic hierarchy process [15]. Furthermore, we define the bandwidth requirement and importance of the $k \in \Pi$ class of services as $F_k$ and $I_k$, respectively, where $\Pi = \{1, 2, ..., K\}$.

Assuming that each service in the power communication network arrives in chronological order, the $n$-th service $s^{(n)}$ can be represented as $s^{(n)} = (v_s^{(n)}, v_d^{(n)}, c^{(n)})$, where $v_s^{(n)}$ and $v_d^{(n)}$ represent the source and destination nodes of the service, and $c^{(n)}$ represents the category of the service.

### 3.3. Network Status

Assuming that the service of the power communication network arrives in chronological order, the current status of network-carrying services forms the basis for service route planning. The status of network-carrying services mainly refers to the status of switching nodes and fiber optic cable links that carry these services. In general, switching nodes and fiber optic links handle various types of services. Therefore, we define the node-carrying service matrix $\boldsymbol{B}_{\mathrm{v}}$ and the link-carrying service matrix $\boldsymbol{B}_{\mathrm{e}}$ to represent the various services carried by nodes and links in the power communication network. The size of $\boldsymbol{B}_{\mathrm{v}}$ is $N \times K$, and its element $\boldsymbol{B}_{\mathrm{v}}(i, k)$ represents the number of $k$-th class services carried by node $v_i$. The size of $\boldsymbol{B}_{\mathrm{e}}$ is $N \times N \times K$, and its elements $\boldsymbol{B}_{\mathrm{e}}(i, j, k)$ and $\boldsymbol{B}_{\mathrm{e}}(j, i, k)$ represent the number of $k$-th class services carried by link $e_{ij}$.

As each service arrives sequentially and its route is deployed, the element values of the matrix will constantly change. Therefore, after the route planning for the $(n-1)$-th service $s^{(n-1)}$ is completed, the node-carrying service matrix and the link-carrying service matrix are $\boldsymbol{B}_{\mathrm{v}}^{(n-1)}$ and $\boldsymbol{B}_{\mathrm{e}}^{(n-1)}$, respectively. Assuming the $n$-th service $s^{(n)}$ arrives, a reachable route for this service is $p$, and the corresponding node-carrying service matrix and link-carrying service matrix for this route are $\boldsymbol{B}_{\mathrm{v},p}^{(n)}$ and $\boldsymbol{B}_{\mathrm{e},p}^{(n)}$, respectively. Assuming that the route planned for this service is $p^{(n)}$, after the deployment of this route is completed, the node-carrying service matrix and link-carrying service matrix are updated to $\boldsymbol{B}_{\mathrm{v}}^{(n)}$ and $\boldsymbol{B}_{\mathrm{e}}^{(n)}$. By defining the node-carrying service matrix and link-carrying service matrix, it is convenient to calculate the network risk and the used bandwidth.

## 4. Problem Description

Based on the power communication network route planning architecture, with the goal of minimizing network risk variance and satisfying the link bandwidth constraint and RPSO constraint, a mathematical description is given for the route allocation planning problem of a single service.

### 4.1. Route Planning Objectives

The network risks primarily consist of the risks associated with switching nodes and fiber optic links. The risk of switching nodes and fiber optic cable links is influenced not only by their own failure rates but also by their significance in carrying services. The higher the failure rate of a node or link itself, or the greater the importance of carrying services, the greater its risk. Based on the definitions of the node-carrying service matrix and the link-carrying service matrix, we can conveniently calculate the network risk.

Assuming a certain reachable route of service $s^{(n)}$ is $p$, the risk values of node $v_i \in V$ and link $e_{ij} \in E$ are, respectively,

$$R_{\mathrm{v},p}^{(n)}(v_i) = r_{\mathrm{v}}(v_i) \cdot g_{\mathrm{v},p}^{(n)}(v_i), \tag{3}$$

$$R_{\mathrm{e},p}^{(n)}(e_{ij}) = r_{\mathrm{e}}(e_{ij}) \cdot g_{\mathrm{e},p}^{(n)}(e_{ij}). \tag{4}$$

Among them, $g_{\mathrm{v},p}^{(n)}(v_i)$ and $g_{\mathrm{e},p}^{(n)}(e_{ij})$ represent the sum of the importance of various services carried by node $v_i$ and link $e_{ij}$, respectively. Based on the definitions of node-carrying service matrix $B_{\mathrm{v},p}^{(n)}$ and link-carrying service matrix $B_{\mathrm{e},p}^{(n)}$, it can be concluded that

$$g_{\mathrm{v},p}^{(n)}(v_i) = \sum_{k \in \Pi} I_k \cdot \boldsymbol{B}_{\mathrm{v},p}^{(n)}(i,k), \tag{5}$$

$$g_{\mathrm{e},p}^{(n)}(e_{ij}) = \sum_{k \in \Pi} I_k \cdot \boldsymbol{B}_{\mathrm{e},p}^{(n)}(i,j,k). \tag{6}$$

The network risk value is the sum of the risk of switching nodes and fiber optic links, i.e.,

$$R_{\mathrm{n},p}^{(n)} = \sum_{v_i \in V} R_{\mathrm{v},p}^{(n)}(v_i) + \sum_{e_{ij}} R_{\mathrm{e},p}^{(n)}(e_{ij}). \tag{7}$$

The average risk value of switching nodes and fiber optic cable links in the power communication network are

$$\overline{R_{\mathrm{v},p}^{(n)}} = \frac{1}{N} \sum_{v_i \in V} R_{\mathrm{v},p}^{(n)}(v_i), \tag{8}$$

$$\overline{R_{\mathrm{e},p}^{(n)}} = \frac{1}{M} \sum_{e_{ij} \in E} R_{\mathrm{e},p}^{(n)}(e_{ij}). \tag{9}$$

The risk variances of switching nodes and fiber optic links are

$$R_{\mathrm{vv},p}^{(n)} = \frac{1}{N} \sum_{v_i \in V} \left( R_{\mathrm{v},p}^{(n)}(v_i) - \overline{R_{\mathrm{v},p}^{(n)}} \right)^2, \tag{10}$$

$$R_{\mathrm{ev},p}^{(n)} = \frac{1}{M} \sum_{e_{ij} \in E} \left( R_{\mathrm{e},p}^{(n)}(e_{ij}) - \overline{R_{\mathrm{e},p}^{(n)}} \right)^2. \tag{11}$$

The definition of network risk variance is the sum of the risk variances of both switching nodes and fiber optic cable links, i.e.,

$$R_{\mathrm{nv},p}^{(n)} = R_{\mathrm{vv},p}^{(n)} + R_{\mathrm{ev},p}^{(n)}. \tag{12}$$

In the power communication network, the smaller the network risk variance $R_{\mathrm{nv},p}^{(n)}$, the smaller the difference in risk values between each switching node and optical cable link, and the better the network operation quality. The variance of network risk is inversely correlated with the quality of network operation. Consequently, minimizing the variance of network risk is set as the objective for power communication network route planning.

*4.2. Route Planning Constraints*

Assuming that a reachable route for service $s^{(n)}$ is $p$, and for link $e_{ij} \in E$, its used bandwidth is

$$w_p^{(n)}(e_{ij}) = \sum_{k \in \Pi} F_k \cdot \boldsymbol{B}_{\mathrm{e},p}^{(n)}(i,j,q). \tag{13}$$

The used bandwidth of a link cannot be greater than the product of link bandwidth capacity and bandwidth utilization threshold, i.e.,

$$w_p^{(n)}(e_{ij}) \leq f(e_{ij}) \cdot \eta_{\mathrm{m}}(e_{ij}). \tag{14}$$

Due to the importance of relay protection services, the route planning process should avoid overburdening any optical cable link with too many relay protection services. The overload threshold for relay protection services is set to $\lambda$, which means that the number of relay protection services carried by each optical cable link in the network cannot exceed $\lambda$. The route planning of the power communication network should be constrained by the overload of relay protection services; that is,

$$\boldsymbol{B}_{\mathrm{e},p}^{(n)}(i,j,1) \leq \lambda, i \in \Lambda, j \in \Lambda. \tag{15}$$

*4.3. Route Planning Problem*

When the *n*-th service $s^{(n)}$ arrives, assuming that the reachable route set of the service is $P^{(n)}$, the route planning problem of the service can be represented as

$$\begin{aligned} \underset{p \in P^{(n)}}{\text{minimize}} \quad & R_{\mathrm{v},p}^{(n)} \\ \text{subject to} \quad & w_p^{(n)}(e_{ij}) \leq f(e_{ij}) \cdot \eta_{\mathrm{m}}(e_{ij}) \\ & \boldsymbol{B}_{\mathrm{e},p}^{(n)}(i,j,1) \leq \lambda, i \in \Lambda, j \in \Lambda. \end{aligned} \tag{16}$$

For relay protection services, it is necessary to carry out dual route planning, comprising a working route $p^{(n)}$ and a protection route $p_{\mathrm{a}}^{(n)}$. When planning the working route, we can follow the description of problem (16). The working and protection routes of a relay protection service must meet the requirement of non-overlapping links; that is, the working route and protection route cannot pass through the same optical cable link. Therefore, the difference between planning the working route of a relay protection service and planning the protection route lies in the set of reachable routes being different, although similar route planning schemes can be adopted.

## 5. Route Planning Scheme

According to question (16), it can be seen that route planning needs to be carried out in two steps. The first step is to find the set of reachable routes given the source and destination nodes of the service. Secondly, in the set of reachable routes, select the route that satisfies the constraint and minimizes the network risk variance as the working route or protection route. However, it is very difficult to directly obtain the set of reachable routes for the service. Therefore, we propose a power communication network route planning scheme based on the SARSA algorithm. Below is a detailed introduction to the SARSA algorithm and its application in route planning.

*5.1. SARSA Algorithm*

The SARSA algorithm is an RL algorithm aimed at maximizing the cumulative benefits of the agent's interaction with the environment, in order to find the optimal strategy. The model of the SARSA algorithm is shown in Figure 2. In this model, the interaction between the agent and the environmental state can be seen as a Markov decision process. The agent can be represented as a quadruple $(S, A, P, R)$, where $S$ is the set of environmental states; $A$ is the set of actions that the agent may take in each state; $P(s_{t+1}|s_t, a_t)$ is the state transition probability model, which represents the probability of taking action $a_t \in A$ to transition to a new state $s_{t+1} \in S$ in state $s_t \in S$; $r_{t+1} = R(s_t, a_t, s_{t+1})$ is the benefit function used to represent the benefits obtained by the agent after taking action $a_t$ in state $s_t$ and transitioning to state $s_{t+1}$.

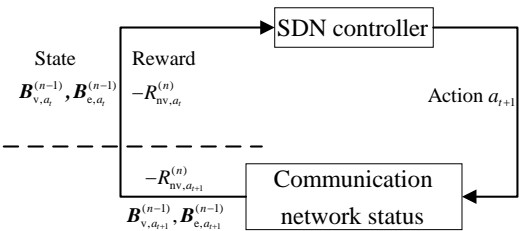

**Figure 2.** SARSA algorithm model diagram.

In the RL algorithm, strategy $\pi : S \to A$ represents the mapping from the state space to the action space. Assuming that the benefits obtained at each future time step must be multiplied by a discount factor $\gamma \in [0,1]$, the sum of benefits from time $t$ to time $T$ is defined as

$$R_t = \sum_{t'=t}^{T} \gamma^{t'-t} r_{t'}. \tag{17}$$

Among them, $\gamma$ is used to measure the impact of future earnings on cumulative earnings.

The state–action function $Q^\pi(s,a)$ refers to executing action $a$ in the current state $s$ and following strategy $\pi$ until the end. The cumulative benefits obtained by the agent during this process are represented as

$$Q^\pi(s,a) = E[R_t | s_t = s, a_t = a, \pi]. \tag{18}$$

For all state–action pairs, if the expected return of a policy $\pi^*$ is greater than or equal to the expected return of other policies, then policy $\pi^*$ is called the optimal policy. There may be more than one optimal strategy, but they share a state–action value function

$$Q^*(s,a) = \max_\pi E[R_t | s_t = s, a_t = a, \pi]. \tag{19}$$

Equation (19) is referred to as the optimal state–action value function, and the optimal state–action value function follows the Bellman optimal equation, i.e.,

$$Q^*(s,a) = E_{s' \sim S}[r + \gamma \max_{a'} Q(s', a') | s, a]. \tag{20}$$

In RL algorithms, the Q-value function is generally solved by iterating the Bellman equation, i.e.,

$$Q_{i+1}(s,a) = E_{s' \sim S}[r + \gamma \max_{a'} Q_i(s', a') | s, a]. \tag{21}$$

Among them, when $i \to \infty$, $Q_i \to Q^*$. By continuously iterating, the state–action value function will converge, resulting in the optimal strategy $\pi^* = \arg\max_{a \in A} Q^*(s,a)$. However, in a large state, the computational cost of using the iterative Bellman equation to solve the Q-value function is too high. Therefore, linear function approximators are commonly used to approximate the state value function.

The SARSA algorithm adopts the Q-value iteration method. In the SARSA algorithm, when taking action $a_t$ in state $s_t$ and changing to state $s_{t+1}$, the calculation expression for $Q(s_t, a_t)$ is

$$Q(s_t, a_t) = (1 - \alpha) Q(s_t, a_t) + \alpha(r_{t+1} + \gamma Q(s_{t+1}, a_{t+1})). \tag{22}$$

Among them, $\alpha$ is the learning factor, which determines the proportion of new information covering old information.

In order to continuously explore new states, the $\varepsilon$-greedy algorithm is generally used for action selection. The specific process is as follows: in a certain state, if the generated random number is less than the greedy rate $\varepsilon$, the next action is randomly selected; otherwise, the action with the highest Q-value is chosen.

In general, the states in a set of states are reachable. Therefore, after a limited number of action selections, it is certain that the target state can be reached from the initial state. But if there are unreachable situations between states, action selection may continue and enter a dead cycle. To solve such problems, we set the upper limit of the number of actions $t_m$, assuming that during the process of training the $Q$-table or obtaining the optimal solution based on the $Q$-table, the number of actions is $t$. When $t = t_m$, if the destination state has not yet been reached, the process ends to avoid entering a dead cycle.

*5.2. Route Planning Algorithm*

To solve problem (16), we propose a power service route planning algorithm based on the SARSA algorithm. Below are the specific details of the algorithm. The proposed route planning algorithm is summarized in Algorithm 1.

In this route planning algorithm, a switching node represents a state, and all nodes form a set of states, i.e., $S = V$.

In state $v_i$, the action is to select a neighboring node of that node as the next node for the route; therefore, the set of actions in this state is $A_i = \{v_j | e_{ij} \in E \text{ or } e_{ji} \in E\}$.

For the benefit function, $R$, if action $a_t$ reaches the destination node, the benefit value is a larger number $\mu$, which enables the algorithm to reach the destination node as soon as possible, i.e., $r_{t+1} = \mu$; otherwise, we calculate the network risk variance $R_{\mathrm{nv},a_t}^{(n)}$ to execute the action and use the opposite of this variance value as the return value, i.e., $r_{t+1} = -R_{\mathrm{nv},a_t}^{(n)}$.

Assuming the maximum number of iterations of the algorithm is $\sigma_m$, when the number of iterations $\sigma$ reaches $\sigma_m$, the algorithm training ends. After the algorithm training is completed, we need to obtain the route based on the $Q$-table. The specific process is as follows: starting from the service source node, we select the action with the highest $Q$-value in each state experienced, and reach the destination node directly to obtain the service route.

---

**Algorithm 1:** Route planning algorithm for the power communication network based on the SARSA algorithm.

---

**Input:** $s^{(n)}$, $\boldsymbol{B}_{\mathrm{v}}^{(n-1)}$, $\boldsymbol{B}_{\mathrm{e}}^{(n-1)}$
**Output:** $p^{(n)}$
1: Initialize the $Q$-table.
2: For $\sigma = 1$ to $\sigma_m$ do
3:     Set $t = 1$, $s_t = v_s^{(n)}$.
4:     While $s_t \neq v_d^{(n)}$ do
5:         In state $s_t$, use the $\varepsilon$-greedy algorithm to select action $a_t$ and enter state $s_{t+1}$.
6:         if $s_{t+1} = v_d^{(n)}$
7:             $Q(s_t, a_t) = (1 - \alpha)Q(s_t, a_t) + \alpha\mu$.
8:             end while
9:         else
10:             Calculate network risk variance $R_{\mathrm{nv},a_t}^{(n)}$, $r_{t+1} = -R_{\mathrm{nv},a_t}^{(n)}$.
11:             $Q(s_t, a_t) = (1 - \alpha)Q(s_t, a_t) + \alpha\left(r_{t+1} + \gamma\max\limits_a Q(s_{t+1}, a)\right)$.
12:             $t = t + 1$.
13:             if $t == t_m$
14:                 end while
15:             end if
16:         end if
17:     end while
18: end for
19: Calculate service route $p^{(n)}$ based on $Q$-table.

---

Due to constraints like link bandwidth and the overload of relay protection services, as the number of services increases, some states are inaccessible. This requires setting the upper limit of the number of actions $t_m$. Assume that during the process of training the $Q$-table or obtaining routes based on the $Q$-table, if the number of actions $t$ reaches $t_m$ but still does not reach the destination node, the process ends. If the process is to obtain the route, it is considered a failure in route planning. When the number of services is small, service route planning is generally not constrained by link bandwidth and the overload of relay protection services. It can be reached between various nodes in the network. In order to fully explore all states, the upper limit of the number of actions can be larger at this time. When there is a large number of services, some nodes cannot be reached. In order to save computing resources, exploration can be terminated in advance, and the upper limit of the number of actions can be reduced. In short, the upper limit of action count should decrease as the number of services increases. In addition, if the action set in a certain state is empty, the current iteration will be terminated directly or route planning will be considered as a failure.

### 5.3. Route Planning Process

Assuming that the power service arrives in chronological order, after completing the route planning of the service $s^{(n)}$, it is necessary to update the node-carrying service matrix $B_v^{(n)}$ and the link-carrying service matrix $B_e^{(n)}$. The flowchart of the power communication network route planning is shown in Figure 3.

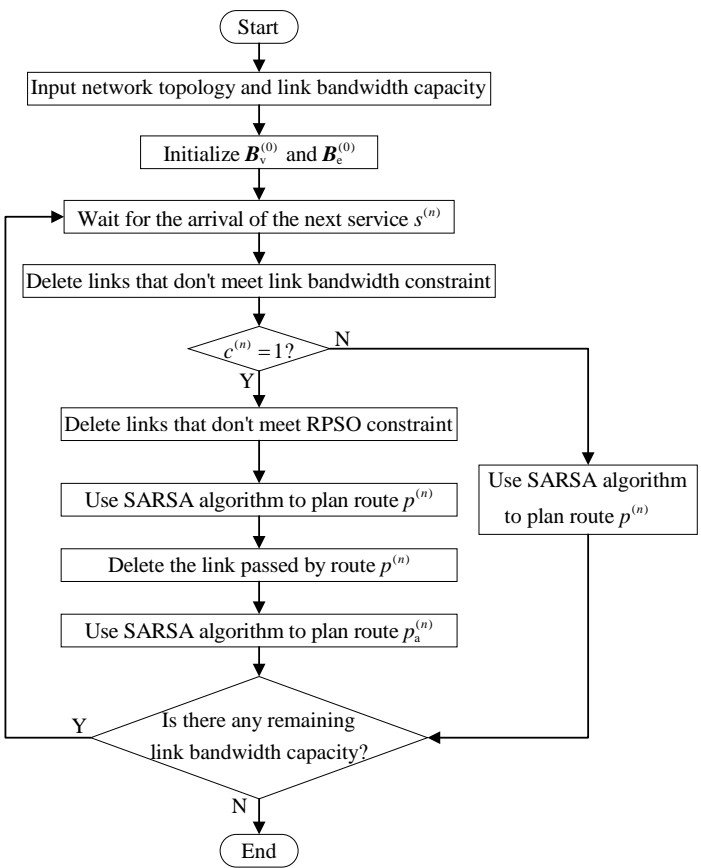

**Figure 3.** Power communication network route planning flowchart.

### 5.4. Evaluating Indicator

According to the definitions of network risk and network risk variance in Section 4.1, in simulation experiments, we mainly use network risk variance to evaluate the degree of network risk equilibrium.

Given the network topology of power communication and various parameters of nodes and links, due to the constraints of link bandwidth capacity and the overload of the relay protection service, the number of services that the network can carry is limited. This may lead to some service route planning failures, resulting in service congestion. In simulation experiments, the performance analysis and research of route planning strategies are carried out using the service blocking rate. Among $\Omega$ services, if the number of services in the $k$-th class is $\Omega_1$ and the number of blocks is $\xi_k$, then the blocking rate of this class of services is $b_k = \xi_k / \Omega_k$, and the total blocking rate is

$$b = \frac{\sum\limits_{k \in \Pi} \xi_k}{\Omega}. \tag{23}$$

Specifically, for non-relay protection services, if the working route planning fails, the service will be blocked; for relay protection services, if the planning of the working route or protection route fails, the service will be blocked.

In addition, the optical cable links have overload constraints for relay protection services. Given that the number of optical cable links is $M$, the link overload rate $o$ is defined as the ratio of the number of overloaded links $M'$ to the total number of links, i.e., $o = M'/M$.

## 6. Simulation Analysis

### 6.1. Simulation Settings

We use the power communication network in certain areas of Shandong Province, China, for experimental simulation to study the route planning scheme. The topology diagram of the communication network is shown in Figure 4, which includes 30 switching nodes and 45 fiber optic links. The link length and number of fiber cores are indicated on the link segments. For example, the optical cable between node $v_1$ and node $v_2$ is 38.2 km, with 36 fiber cores. In practice, the bandwidth capacity of fiber optic cable links is directly proportional to the number of fiber cores. For the convenience of the experiment, we will set the bandwidth of the optical cable link in proportion to the actual number of fiber cores.

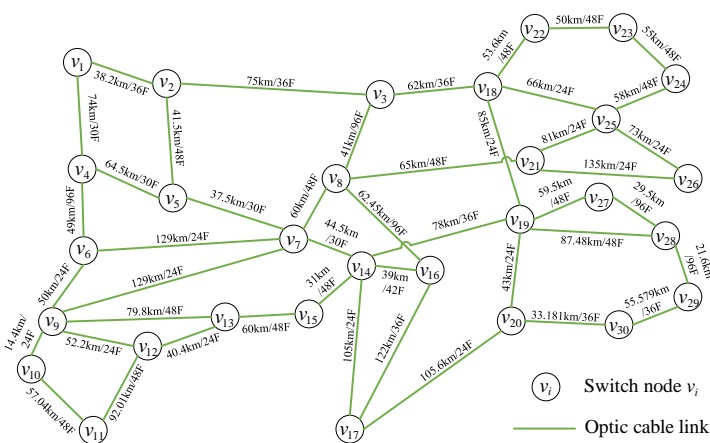

**Figure 4.** Topology diagram of the power communication network in certain areas of Shandong Province, China.

According to Section 5.2, the upper limit of the number of actions $t_m$ decreases with the increase in the number of transactions $\Omega$. Therefore, when $\Omega \leq 100$, we set $t_m = 1000$, and when $\Omega > 100$, we set $t_m = 100$. Other simulation parameters are shown in Table 1.

**Table 1.** Simulation parameters and parameter values.

| Parameter | Value | Parameter | Value |
|:---:|:---:|:---:|:---:|
| $u_v(v_i)$ | 0.9996 | $\alpha$ | 0.1 |
| $u_e(e_{ij})$ | 0.9984 | $\gamma$ | 0.9 |
| $\eta_m(e_{ij})$ | 1 | $\varepsilon$ | 0.1 |
| $\lambda$ | 8 | $\sigma_m$ | 100 |

In practice, power services are mainly divided into six categories, and the bandwidth requirements, importance, and number of each type of service are shown in Table 2. In the simulation, we randomly generate various types of services based on their proportional quantities. The source and destination nodes of non-relay protection services are randomly selected from the node set according to a uniform distribution. The source and destination nodes of relay protection services are adjacent and directly connected by links.

**Table 2.** Service types and related indicator data.

| Service Type | Bandwidth Demand [Mbits/s] | Importance | Quantity Proportion |
|:---:|:---:|:---:|:---:|
| Relay protection service | 2 | 0.9981 | 5 |
| Stably control system service | 2 | 0.6069 | 10 |
| Schedule automation service | 2 | 0.1008 | 20 |
| Communication monitoring service | 2 | 0.0768 | 15 |
| Management telephone service | 0.5 | 0.0652 | 30 |
| Information support system service | 10 | 0.0234 | 20 |

Considering the randomness of service generation, sufficient samples are needed for experimentation. In the simulation, it is assumed that there are multiple services in a service group. Assuming that the services arrive in chronological order, route planning is carried out for the services in the service group in sequence until all service route planning is completed. After completing all service route planning in the service group, we calculate the relevant simulation results, such as the service blocking rate and network risk variance. Furthermore, we generate 100 service groups with the same number of services. We average the results of multiple service groups as the final simulation result.

*6.2. Method Comparison*

To verify the effectiveness of the proposed scheme, a comparative analysis is conducted with other existing route planning schemes. For ease of description, the scheme proposed in this article is referred to as SARSARoute. Reference [13] proposes a route planning algorithm that considers a joint balance of the link load and service risk. In this study, the authors assess the risk and load of the link, using their weighted average as the link weight. They then employ the Dijkstra algorithm to determine the path with the minimum weight value as the service route. This algorithm is denoted as LRJB, with a balance factor of 0.45 in the algorithm. Reference [14] proposes a route planning algorithm that constrains network risk values, denoted as RiskRoute. In this reference, the authors consider both link risk and node risk. The risk value of the route is defined as the sum of the link and node risk values. The authors use an improved Dijkstra algorithm to obtain the path with the lowest risk value as the service route.

In terms of the algorithm's time complexity, both the LRJB algorithm and RiskRoute algorithm are based on the Dijkstra algorithm, with a time complexity of $O(N^2)$, where $N$ is the number of nodes. Assuming the number of training rounds for the SARSARoute algorithm is $\sigma_m$, and the upper limit of action selections during one training round is $t_m$, the time complexity of the SARSARoute algorithm is $O(\sigma_m t_m)$. $t_m$ is the upper limit of

action selections. As the number of nodes $N$ increases, $t_m$ also increases, allowing for a more comprehensive exploration of the state space.

We compared and analyzed the above schemes; the results are as follows.

### 6.2.1. Comparison between the Service Blocking Rate and Link Overload Rate

Figures 5 and 6 show the comparison between the service blocking rate $b$ and the relay protection service blocking rate $b_1$ for various schemes under different quantities of services $\Omega$. Figure 7 compares the link overload rate $o$ for various schemes under different quantities of services $\Omega$. From Figures 5 and 6, it can be seen that both the service blocking rate $b$ and the relay protection service blocking rate $b_1$ gradually increase with the increase in the number of services $\Omega$, and the service blocking rate $b$ and relay protection service blocking rate $b_1$ of the SARSARoute route planning scheme are higher than the other two schemes. From Figure 7, it can be seen that the link overload rate $o$ of the SARSARoute route planning scheme has always been 0, meaning no links experience relay protection service overload, while the link overload rate $o$ of the other two schemes will gradually increase with the increase in service number $\Omega$. The reasons are as follows.

The SARSARoute route planning scheme accounts for the overload constraint of the relay protection service. The number of relay protection services carried by an optical cable link is restricted not only by the bandwidth capacity but also by the overload constraint of the relay protection service. Therefore, the relay protection service blocking rate of the SARSARoute route planning scheme is relatively high, which further leads to a higher service blocking rate. Therefore, using the SARSARoute route planning scheme, all optical cable links will not experience overload, effectively improving the reliability of relay protection services.

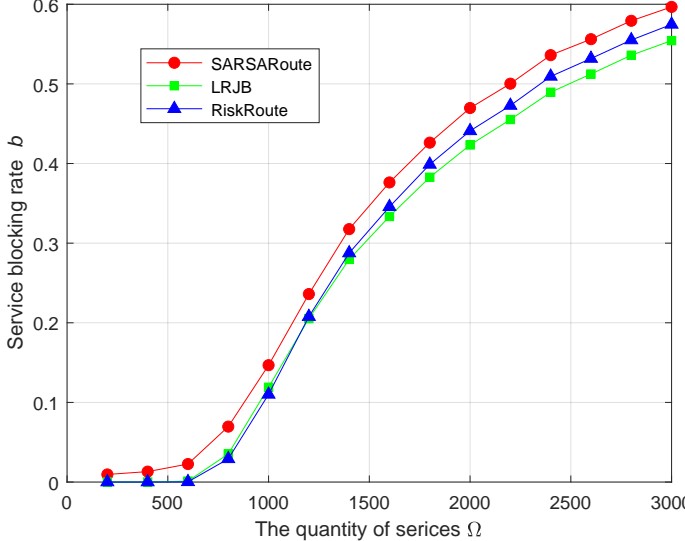

**Figure 5.** Comparison of service blocking rates $b$ for various schemes under different quantities of services $\Omega$.

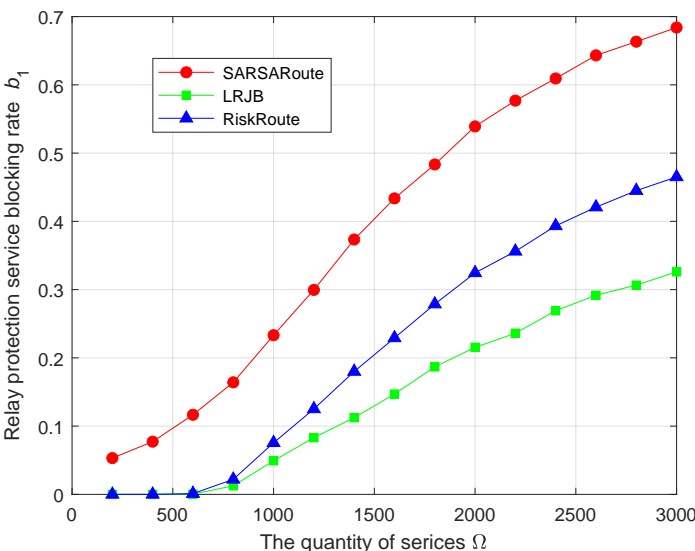

**Figure 6.** Comparison of relay protection service blocking rates $b_1$ for various schemes under different quantities of services $\Omega$.

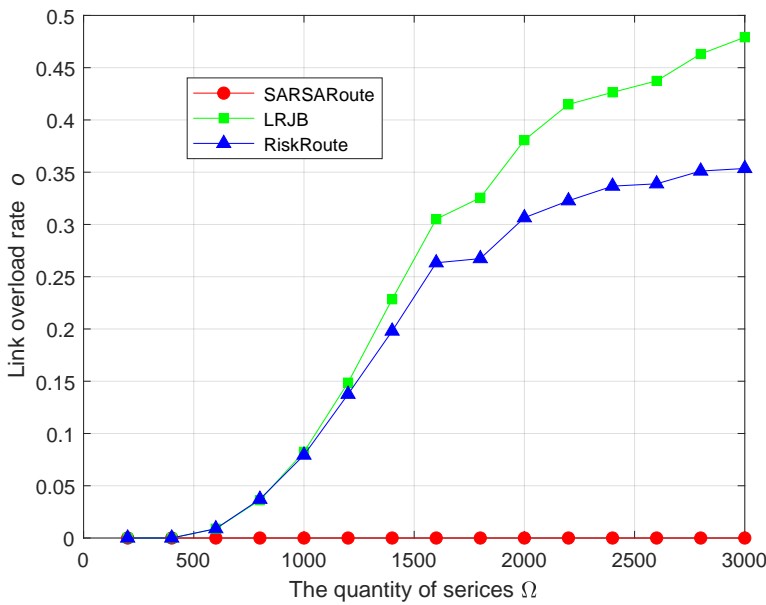

**Figure 7.** Comparison of link overload rates $o$ for various schemes under different quantities of services $\Omega$.

### 6.2.2. Comparison of Network Risk

Figures 8 and 9 show the comparison of the network risk value $R_n^{(n)}$ and network risk variance $R_{nv}^{(n)}$ for various schemes under different quantities of services $\Omega$. From Figures 8 and 9, it can be seen that the network risk value $R_n^{(n)}$ and network risk variance $R_{nv}^{(n)}$ both increase with the increase in the number of services $\Omega$, but the network risk value $R_n^{(n)}$ and network risk variance $R_{nv}^{(n)}$ of the SARSARoute route planning scheme are smaller than the other two route planning schemes. The reasons are as follows.

For the network risk value $R_n^{(n)}$, we know that the importance of the relay protection service is the highest, and the corresponding risk value is relatively high. Compared to the other two schemes, the SARSARoute route planning scheme considers the overload constraint of relay protection services, and some relay protection services are blocked. In addition, the LRJB scheme does not consider node risk. Therefore, the network risk

value $R_{\mathrm{n}}^{(n)}$ of the SARSARoute route planning scheme is minimized. For the network risk variance $R_{\mathrm{nv}}^{(n)}$, the SARSARoute route planning scheme obtains the route with the smallest network risk variance through trial and error. But the LRJB scheme considers the load balance and risk balance of the link, without considering the risk of nodes. The purpose of the RiskRoute scheme is to find the route with the lowest risk, rather than the route that minimizes the network risk variance. Therefore, the network risk variance $R_{\mathrm{nv}}^{(n)}$ of the SARSARoute route planning scheme is minimized.

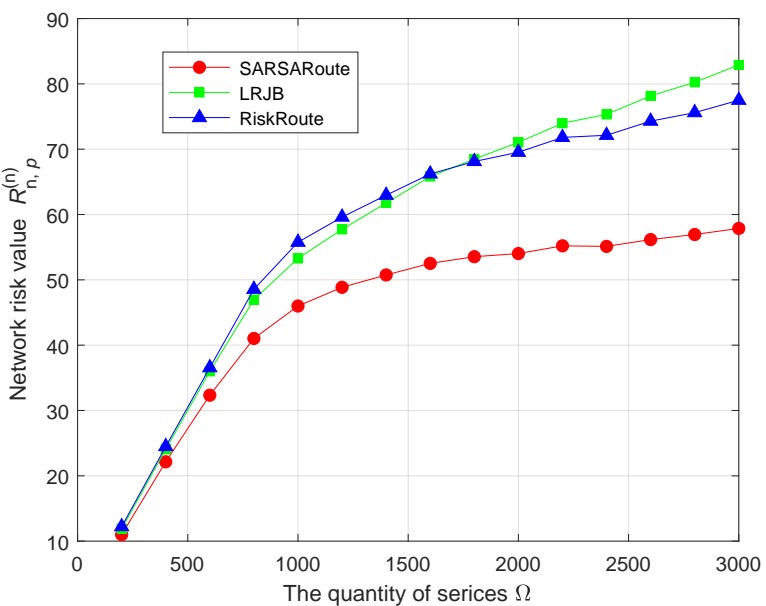

**Figure 8.** Comparison of network risk values $R_{\mathrm{n}}^{(n)}$ for various schemes under different quantities of services $\Omega$.

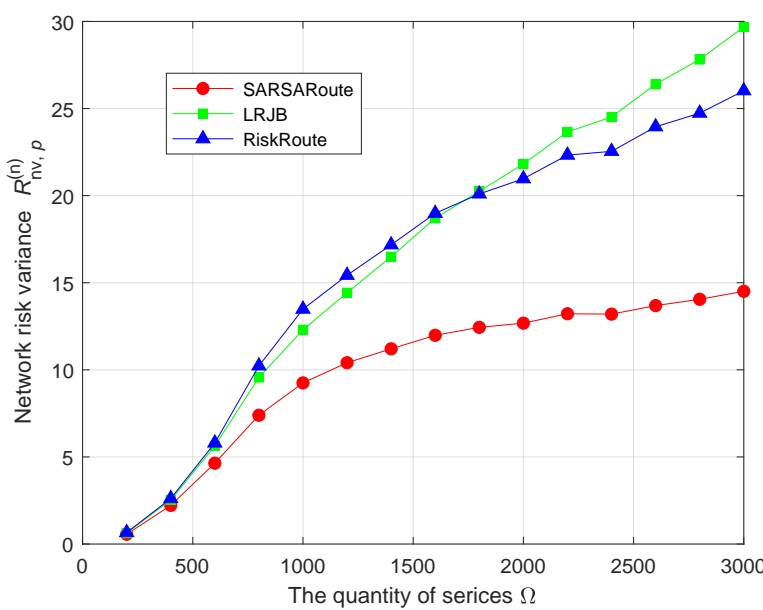

**Figure 9.** Comparison of network risk variance $R_{\mathrm{nv}}^{(n)}$ for various schemes under different quantities of services $\Omega$.

### 6.2.3. Summary of Methods: Comparison

Table 3 shows the performance comparison of different schemes on the power communication network in areas of Shandong Province, China. According to Table 3, the LRJB

scheme performs the best in terms of the service blocking rate. Based on Figure 5, it can be seen that the SARSARoute scheme is not significantly different from this scheme. However, the SARSARoute scheme is superior to the other two schemes in terms of the link overload rate and network risk. Therefore, the scheme proposed in this paper is capable of avoiding the overloading of optical cable links, effectively reducing the variance of network risk, and balancing network risk.

**Table 3.** Performance comparison of various schemes on the power communication network in certain areas of Shandong Province, China.

| Schemes | Service Blocking Rates $b$ | Link Overload Rates $o$ | Network Risk Values $R_{\text{n}}^{(n)}$ | Network Risk Variance $R_{\text{nv}}^{(n)}$ |
|---|---|---|---|---|
| SARSARoute | | ✓ | ✓ | ✓ |
| LRJB | ✓ | | | |
| RiskRoute | | | | |

✓ indicates that the performance of this scheme is the best.

### 6.3. Simulation Results on the Robustness of the Service Route Planning Scheme

To further verify the effectiveness and study the robustness of the proposed scheme in this paper, the NSFNet network is selected as the power communication network. As shown in Figure 10, this network consists of 14 nodes and 21 links. The length of each link is randomly selected in the range of 50–150 km, and the bandwidth capacity is set to 100 Mbits/s. Other simulation parameters remain unchanged. Similar to Section 6.2, we study the service blocking rate, link overload rate, network risk value, and network risk variance of three schemes under different quantities of services. We summarize the simulation results and place them in Table 4. According to Table 4, the SARSARoute scheme is superior to the other two schemes in terms of the link overload rate and network risk. The results are consistent with the comparison results in Table 3. Therefore, the proposed scheme in this paper is capable of avoiding the overloading of relay protection services and balance network risks in different network topologies.

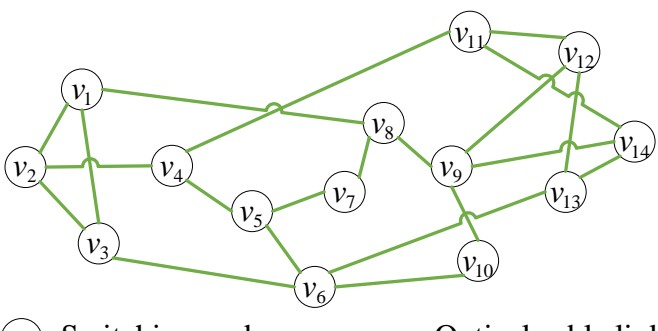

$v_i$ Switching node $v_i$ ——— Optical cable link

**Figure 10.** Topology diagram of NSFNet.

**Table 4.** Performance comparison of various schemes on NSFNet.

| Schemes | Service Blocking Rates $b$ | Link Overload Rates $o$ | Network Risk Values $R_{\text{n}}^{(n)}$ | Network Risk Variance $R_{\text{nv}}^{(n)}$ |
|---|---|---|---|---|
| SARSARoute | | ✓ | ✓ | ✓ |
| LRJB | ✓ | | | |
| RiskRoute | | | | |

✓ indicates that the performance of this scheme is the best.

## 7. Conclusions

In order to plan power service routes reasonably and improve the reliability of power service transmission, we propose a power service route planning scheme based on the SARSA algorithm. Firstly, we establish a power service route planning architecture based on SDN and define power service and network status parameters. On this basis, the route planning problem is characterized by the goal of minimizing network risk variance and adhering to constraints on the link bandwidth and relay protection service overload. To tackle this issue, a power service route planning scheme based on the SARSA algorithm is proposed. In the simulation, we evaluate the performance of the proposed method by comparing it with existing methods. The simulation results show that the proposed route planning scheme is superior to other solutions in terms of relay protection service overload and network risk. The proposed route planning scheme can effectively avoid the overload of relay protection service, balance network risk, and improve the reliability of power communication network service transmission.

In the future, we will consider building a more comprehensive service route planning model. On the one hand, at the end of the power communication network, there is generally a wireless access network. Combining the communication backbone network with the wireless access network can create a more extensive model. On the other hand, with the development of the smart grid, the variety and number of services will continue to increase. Different types of services have different route planning requirements. To address more complex system models and route planning issues, we will consider using more advanced algorithms to complete service route planning, such as deep reinforcement learning.

**Author Contributions:** Conceptualization, X.L. (Xinquan Lv) and Y.W.; methodology, X.L. (Xiaolong Liu); validation, X.L. (Xiaolong Liu) and P.M.; investigation, K.M. and Y.Z.; resources, C.S.; writing—original draft preparation, X.L. (Xiaolong Liu); writing—review and editing, P.M. All authors have read and agreed to the published version of the manuscript.

**Funding:** This work was supported by the Science and Technology Project of the State Grid Corporation of China (Research on Dispatching Fusion Communication Oriented to Power Communication Network and Its Cooperative Control with Power Network Operation, 52060022001B).

**Data Availability Statement:** Data is contained within the article.

**Conflicts of Interest:** Author Xinquan Lv, Yongjing Wei, Kai Ma, Chao Sun and Youxiang Zhu were employed by the company State Grid Shandong Electric Power Company. The remaining authors declare that the research was conducted in the absence of any commercial or financial relationships that could be construed as a potential conflict of interest.

## Abbreviations

The following abbreviations are used in this manuscript:

| | |
|---|---|
| SDN | software-defined network |
| RL | reinforcement learning |
| DL | deep learning |
| RPSO | relay protection service overload |
| SARSA | state–action–reward–state–action |
| KSP | K shortest path |

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
