# Peer review of "Research on Power Service Route Planning Scheme Based on SDN Architecture and Reinforcement Learning Algorithm"

_electronics, doi:10.3390/electronics13020386_

Round 1

Reviewer 1 Report

Comments and Suggestions for Authors

1. This paper propose a route planning architecture for power communication network based on SARSA algorithm. 

For showing the data flow of SARSA algorithm, I think Figure 2 would be modified in detail.

2. Table 1 shows simulation parameters and values and Table 2 shows service types and related indicator data. I think it would be better for the author to present more simulation cases for showing the performance of proposed algorithms.

Comments on the Quality of English Language

The English expression is considered appropriate, but the algorithm explanation such as SARSA part needs to be reviewed more clearly.

Reviewer 2 Report

Comments and Suggestions for Authors

The paper is well -written; however, there are several concerns that are needed to be addressed as following:

1) The reviewer suggest to highlight the reasons to select RL over supervised or unsupervised models? 

2) what is the main contribution of this study over otter study in literature 

3) The reviewed suggests to add a table of comparison at the end of the results 

4) please modify abstract and conclusion 

5)Please check the errors and grammar check 

Comments on the Quality of English Language

The quality is fine . Please review the paper again for improvements. 

Reviewer 3 Report

Comments and Suggestions for Authors

In their study, the researchers developed a network routing architecture that integrates Software Defined Network (SDN) principles with the SARSA reinforcement learning algorithm. This novel approach is primarily focused on reducing network risk variability, taking into account critical factors such as link bandwidth utilization. The effectiveness of this methodology is validated through simulations, which highlight its proficiency in mitigating network overloads, equilibrating network risk, and bolstering data transmission dependability within power communication networks.

The paper contrasts the performance of three distinct algorithms: SARSA, LRJB, and RiskRoute. Notably, reference 12, titled "An Optimized Routing Algorithm with Load and Risk Joint Balance in Electric Communication Network," is anticipated to provide in-depth insights into the LRJB algorithm. However, this paper was not locatable in the CSEE 2018 proceedings, suggesting a need for more comprehensive elucidation of the LRJB algorithm. Meanwhile, reference 13 details the RiskRoute method, which is an adaptation of the Dijkstra algorithm.

In the introductory section, it was highlighted that Q-learning and SARSA are among the predominant methods in reinforcement learning. A comparative analysis of their performance is warranted, given that both methods represent distinct strategic approaches within the reinforcement learning framework.

Furthermore, a noteworthy parallel is drawn with another research piece employing the SARSA method for route optimization in power service networks (Zhang, Geng, et al., "A Service Routing Optimization Algorithm for Power Communication Optical Transport Network Based on Knowledge Graph and Reinforcement Learning," International Conference on Autonomous Unmanned Systems, Springer Singapore, 2021). A comparative analysis of the innovative aspects of the current study against this existing research could provide valuable insights into the advancements in the field.

In the citations section, it is crucial to maintain consistency in formatting. For instance, discrepancies are noted where some citations have bolded the year of publication, while others have not. To ensure uniformity, it would be advisable to apply a consistent style to all citations, either by uniformly bolding the year in all references or refraining from this practice altogether.

Round 2

Reviewer 2 Report

Comments and Suggestions for Authors

Thank you for addressing the comments. Please double check the English errors before publication

Comments on the Quality of English Language

Please double check the grammar.

Reviewer 3 Report

Comments and Suggestions for Authors

While the author has responded to a significant portion of the previous review questions, certain clarifications are still required.

It would be beneficial for the manuscript if the LRJB method, used as the comparative approach, is briefly explained rather than solely relying on a citation.

Given that Citation 25 has already conducted a comparison between the Sarsa and Q-learning methods in the context of power service route planning, what novel contributions does this paper bring to the table when juxtaposed with the Sarsa method mentioned in the citation?

In light of the convergence challenges faced by Sarsa which will affect system robustness, attributed to factors such as environmental complexity, the delicate balance between exploration and exploitation, function approximation nuances, and the intricacies of hyperparameter tuning, how can we strategically navigate these issues to prevent convergence pitfalls?

Have we completed any computations pertaining to the time-complexity comparison?

Round 3

Reviewer 3 Report

Comments and Suggestions for Authors

The author addressed all of my inquiries, no more questions.